# Aberrant BMP2 Signaling in Patients Diagnosed with Osteoporosis

**DOI:** 10.3390/ijms21186909

**Published:** 2020-09-21

**Authors:** Hilary W. Durbano, Daniel Halloran, John Nguyen, Victoria Stone, Sean McTague, Mark Eskander, Anja Nohe

**Affiliations:** 1Department of Biological Sciences, University of Delaware, Newark, DE 19716, USA; weidnerh@udel.edu (H.W.D.); dhallor@udel.edu (D.H.); njohn@udel.edu (J.N.); vstone@udel.edu (V.S.); 2Christiana Care Hospital, Newark, DE 19716, USA; smctague@christianacare.org (S.M.); markeskander77@gmail.com (M.E.)

**Keywords:** osteoporosis, BMP2, CK2.3, BMPRIa, CK2

## Abstract

The most common bone disease in humans is osteoporosis (OP). Current therapeutics targeting OP have several negative side effects. Bone morphogenetic protein 2 (BMP2) is a potent growth factor that is known to activate both osteoblasts and osteoclasts. It completes these actions through both SMAD-dependent and SMAD-independent signaling. A novel interaction between the BMP type Ia receptor (BMPRIa) and casein kinase II (CK2) was discovered, and several CK2 phosphorylation sites were identified. A corresponding blocking peptide (named CK2.3) was designed to further elucidate the phosphorylation site’s function. Previously, CK2.3 demonstrated an increased osteoblast activity and decreased osteoclast activity in a variety of animal models, cell lines, and isolated human osteoblasts. It is hypothesized that CK2.3 completes these actions through the BMP signaling pathway. Furthermore, it was recently discovered that BMP2 did not elicit an osteogenic response in osteoblasts from patients diagnosed with OP, while CK2.3 did. In this study, we explore where in the BMP pathway the signaling disparity or defect lies in those diagnosed with OP. We found that osteoblasts isolated from patients diagnosed with OP did not activate SMAD or ERK signaling after BMP2 stimulation. When OP osteoblasts were stimulated with BMP2, both BMPRIa and CK2 expression significantly decreased. This indicates a major disparity within the BMP signaling pathway in patients diagnosed with osteoporosis.

## 1. Introduction

Osteoporosis (OP) is a debilitating bone disease affecting approximately one in three women and one in five men age 50 years and older, worldwide [1]. As the aging population increases, the number of patients who will be diagnosed with OP will also increase [2,3]. Therefore, there is an urgent need to further delineate the causes of OP and develop novel therapeutics to treat this disease. Currently, there are six different types of osteoporotic medications clinically available. A large majority of these treatments are antiresorptive, meaning they focus on decreasing bone resorption. There are very few treatments on the market that are anabolic, which increase bone formation [4]. Currently, there is one available treatment, namely Romosozumab, that increases bone formation while decreasing bone resorption [5]. However, because of its recent approval from the Food and Drug Administration (FDA), its long-term side effects have not been well-characterized [6,7]. Therefore, alternate therapeutics need to be developed, but in order to create more effective drug options, bone maintenance and homeostasis must be understood.

The bones are maintained and remodeled through the bone remodeling system. There are two major cell types in this system, which include osteoclasts and osteoblasts. Osteoclasts are responsible for resorbing back old or damaged bone, while osteoblasts form new bone [8]. In a normal, healthy individual, there is a balance between osteoclast activity and osteoblast activity. However, in OP patients, there is an imbalanced relationship between these two cell types, causing increased osteoclast activity and decreased osteoblast activity [9]. This leads to increased porosity in bone, which increases the risk of fractures [9]. In 2015, there were 2.3 million osteoporotic fractures reported in the United States; therefore, understanding this disease and developing more viable treatments that target both osteoblasts and osteoclasts is of the utmost importance [1].

Bone morphogenetic protein 2 (BMP2) is a potent growth factor that is known to induce both osteoblast activity and osteoclast activity [10]. BMP2 has been approved by the FDA for the healing of long bone fractures and spinal fusion surgeries [11]. Its potential use as an osteoporotic treatment has been explored; however, long-term use of BMP2 has been linked to increased osteoclast activity and bone resorption [12]. In addition, there are a multitude of negative side-effects following the use of BMP2 in the clinic, such as vertebral osteolysis, hematoma and seroma formation [11]. Therefore, while BMP2 itself is not an optimal treatment for OP, studying and analyzing the BMP pathway will elucidate the mechanism of osteoblast and osteoclast activation, and this pathway may be implicated for future therapeutics.

Interestingly, several research groups have discovered a potential signaling disparity within the BMP pathway with those diagnosed with OP [13,14]. Further, we have previously reported that osteoblasts extracted from patients diagnosed with OP did not respond to BMP2 stimulation through mineralization assessments [15]. Mineralization deposits consist of mainly calcium and phosphate, thus indicating whether BMP2 is stimulating bone formation in OP patients when compared to control patients. Additionally, it has been demonstrated that there is a decrease in BMP2 induced SMAD signaling (the canonical signaling pathway) when compared to control patients, as indicated by decreased SMAD4 levels [13].

BMP2 can bind to two receptors responsible for initiating the canonical and non-canonical signaling cascades, which are known as the type I receptor (BMPRIa) and the type II receptor (BMPRII). BMP2 induced signaling is initiated when the protein either recruits the binding of the two receptors to dimerize or binds to already dimerized receptors [16]. Once all three components are bound together in a complex, the type II receptor will phosphorylate the type I receptor within its glycine-serine-rich (GS) box homeodomain [17]. Subsequently, both SMAD independent and SMAD dependent signaling may be activated [17].

Our lab previously discovered an interacting protein called casein kinase 2 (CK2) that is released upon BMP-2 activation. A prosite search revealed potential phosphorylation sites on BMPRIa that CK2 could bind. We previously designed a peptide sequence that mimics a BMPRIa phosphorylation site, with residues flanking on either side homologous with CK2 in order to promote peptide and CK2 binding [16,18,19]. This peptide (CK2.3) is designed to bind to CK2 and prohibit its interaction with the BMPRIa receptor at the corresponding phosphorylation site, but not to inhibit CK2 from binding to BMPRIa completely. CK2.3 has been shown to increase bone formation and decrease bone resorption in various animal models [15,20,21,22]. It has also been shown to elicit its response primarily through the ERK pathway, and the SMAD pathway to a lesser extent [22]. CK2.3 also increased mineralization in OP patients, when BMP2 did not, showing the increased potential of CK2.3 as a novel OP therapeutic, especially if there is a mis-regulation within the canonical BMP signaling cascade [15].

Additionally, Donoso and colleagues discovered that there was an overexpression of BMPRIa in mesenchymal stem cells (MSCs) extracted from the bone marrow of OP patients [13]. We hypothesize that this newly discovered signaling disparity is through receptor dysregulation due to increased levels of BMPRIa in OP patients. This overexpression is indicative of a signaling bias in the BMP pathway and it is important to further investigate receptor expression in OP patients. In this paper, we investigate the possibility that BMP2 is not inducing both SMAD dependent and SMAD independent signaling (ERK). We looked solely at SMAD and ERK activation through immunofluorescence and conformational Western blots, since CK2.3 and BMP2 are known to activate both signaling cascades. We also studied expression levels of BMPRIa and CK2 through various immunofluorescent models after BMP2 and CK2.3 stimulation. This was to further elucidate BMP2 and CK2.3′s effect on BMPRIa and CK2 expression in OP patients as several other researchers noted an overexpression within their osteoporotic patient pool. Further analysis within the BMP signaling cascade in OP patients is critical, as this could lead to the development of more effective therapeutics.

## 2. Results

### 2.1. BMP2 Stimulation Decreases pERK Expression in OP Patients

Since BMP2 stimulation did not induce an osteogenic mineralization response, other downstream signaling proteins need to be further investigated in order to determine where the BMP signaling disparity lies. The SMAD independent signaling encompasses a variety of pathways, one of which is the ERK signaling pathway. To analyze the ERK signaling pathway, mature osteoblasts were plated, grown, and stimulated with either BMP2, CK2.3 or left unstimulated (US). After the fifth day, cells were stained immunofluorescently for pERK (green), and the nucleus (blue), Figure 1a. BMP2 stimulation significantly decreased expression of pERK, or activated ERK, when compared to US and CK2.3 stimulated cells. Interestingly, CK2.3 and US cells had significantly increased expression of pERK over BMP2. These results were further validated through an immunoblot. Briefly, patient cells were stimulated with BMP2, CK2.3 or left US for five days. On the fifth day, lysates were collected, and protein concentration was determined and normalized. Lysates were run on an SDS-PAGE gel, transferred to a PDVF membrane, and immunoblotted for pERK and β-actin. BMP2 significantly decreased expression of pERK when compared to CK2.3 stimulated cells, Figure 1b. This indicates ERK/SMAD independent signaling is disrupted in OP when stimulated with BMP2.

### 2.2. SMAD Expression Remains Unchanged in OP Patients

Since ERK signaling was disrupted in OP patients and is part of the SMAD independent signaling pathway, SMAD dependent signaling was also investigated. Canonical SMAD signaling or SMAD dependent signaling is the most studied BMP signaling pathway. Patient cells were stimulated with BMP2, CK2.3, or left US for five days. After the fifth day, cells were immunofluorescently labeled for pSMAD (green) and the nucleus (blue), Figure 2a. While BMP2 stimulation seemed to decrease pSMAD (activated SMAD) expression, it was not significant. These results were again confirmed through a Western blot. The patient cells lysates were prepared as outlined above in Section 2.1. Again, pSMAD expression did not significantly change under all stimulations, Figure 2b. This indicates that the BMP signaling disparity was not observed within the SMAD dependent pathway.

### 2.3. Trabecular Bone Explants as a Viable Bone Model

We wanted to further explore how the native bone tissue itself reacted to stimulations with BMP2 and CK2.3. Therefore, we utilized an explant model to observe these effects. Explant models are beneficial models because they utilize the organ of interest, allowing for investigation of the stimulations or treatments used. To prepare our samples for an explant study, the femoral heads from both control and OP patients were cut down the midsagittal plane and the area of interest is indicated in Figure 3a. Bone fragments were then fixed and stained for Calcein red-orange and Hoescht in order to determine efficacy and viability of the cells within the bone, as well as the bone explant model, Figure 3c. Calcein stains viable cells, while Hoescht is a nuclear stain that stains both live and dead cells. The bone fragments were imaged using confocal microscopy and the percent viability of the cells were determined. In both control and OP bone fragments, in all stimulations, cell viability was over 80%. This shows the efficacy of the trabecular bone model.

### 2.4. BMP2 Stimulation Decreases OC and ALP Expression in OP Patients Bone Explants When Compared to Control Patients

After the creation of a viable explant model, we wanted to further validate its effectiveness. Previously, we showed that primary osteoblasts isolated from OP patients led to decreased expression of the osteoblast specific markers, namely OC and ALP, when stimulated with BMP2 [15]. Here, we conducted the same study as described by Weidner et al., except we used the explant model and included control patients as a comparison. The explants were stimulated with BMP2, CK2.3, or left US for five days. After the fifth day, the explants were fixed in 4.4% PFA and immunofluorescently labeled for OC (green) and ALP (red), Figure 4a,c. Representative images can be seen in Figure 4b,d, respectively. Again, BMP2 stimulation significantly decreased expression of both OC and ALP in OP patients. However, CK2.3 significantly increased expression when compared to both BMP2 and US. Interestingly, both BMP2 and CK2.3 significantly increased expression of OC and ALP in control patients when compared to US. This shows the validity of the trabecular explant model.

### 2.5. Increased Basal Levels of BMPRIa and CK2α Expression in OP Patients

Human femoral heads were isolated from female patients diagnosed with OP, as well as control. The specimens were preserved in 10% Nuetral Buffered Formalin (NBF) solution within 48 h of extraction. The samples were then cut down the midsagittal plane, and a 2 mm bone fragment was removed and embedded in MMA to protect the integrity of the bone microenvironment. Other embedded methods, such as paraffin embedding, decalcify the bone before embedding, which remove meaningful and important parameters when investigating skeletal based diseases such as OP. The embedded bone fragments were stained for BMPRIa and CK2α, and increased expression levels of both proteins were observed in the OP specimens, Figure 5. Control specimens did have expression of both BMPRIa and CK2α, however, OP patients demonstrated significantly increased expression of both proteins. This further indicates a signaling disparity within the BMP pathway that needs to be investigated.

### 2.6. BMP2 Stimulation Decreases BMPRIa and CK2α Expression in OP Patients Bone Explants

The bone explant model was utilized to study BMPRIa and CK2α expression after BMP2 and CK2.3 stimulation. Femoral heads from both control and OP samples were used and after the fifth day of stimulation, once fixed, the samples were labeled immunofluorescently for BMPRIa (green), Figure 6a,b, and CK2α (red), Figure 6c,d. Representative images can be seen below each figure. In control samples, there was a significant increase in BMPRIa and CK2α after both BMP2 and CK2.3 expression when compared to US. In OP samples, there was a significant decrease in BMPRIa and CK2α expression when compared to US and CK2.3 stimulated samples. This mimicked the response observed in the MMA trabecular bone slices, Figure 3.

### 2.7. BMPRIa and CK2α Levels in Osteoblasts from OP Patients

BMPRIa and CK2α expression levels were further validated in an in vitro model. Mature osteoblasts were isolated from OP patients as previously described. Cells were plated and stimulated with BMP2, CK2.3, or left US. After five days, the cells were fixed and stained for BMPRIa (green), CK2α (red), and nucleus (blue). There was a significant increase in BMPRIa and CK2α expression in CK2.3 stimulated cells when compared to US and BMP2 stimulation, Figure 7a,b. This mimicked the responses we had seen in both MMA embedded bone and the explant models. Finally, RNA was isolated from the cells and gene levels of *BMPRIa* were detected through RT-PCR, Figure 7c. This further indicates a potentially BMP2 induced signaling disruption in OP as it mimics the responses seen in the MMA trabecular bone slices (Figure 5) and the bone explants (Figure 6).

### 2.8. Proposed Mechanisms for BMP2 and CK2.3

In congruence with previous findings, the following mechanisms are proposed for BMP signaling in patients diagnosed with OP. As seen in Figure 8a, when the extracted cells from OP patients are stimulated with BMP2, they are unresponsive in their osteogenic response [23]. They have significantly decreased ERK and SMAD activation, in addition to significantly decreased immunofluorescent expression of BMPRIa, and CK2α. Where the discrepancy within the BMP pathway lies needs to be further investigated. Consequently, CK2.3 does induce mineralization in cells extracted from OP patients [23]. CK2.3 has been shown to activate SMAD and ERK signaling in C2C12 cells lines [22], and now has been shown to increase ERK activation in cells extracted from patients diagnosed with OP, Figure 8b.

## 3. Discussion

Osteoporosis is the most prevalent skeletal disease in humans, and its occurrence is expected to increase with the increasing aging population [2]. Current therapeutics on the market are insufficient because they are either anabolic (increase osteoblast activity) or antiresorptive (decrease osteoclast activity), and there is only one therapeutic that focuses on both, namely Romosozumab. Therefore, there is an urgency to develop new therapeutics that can stimulate both osteoblast and decrease osteoclast activity. One potential therapeutic, BMP2, is a growth factor that induces both osteoblastogenesis and osteoclastogenesis, and therefore, seems to be an ideal candidate for an OP therapeutic. Furthermore, BMP2 has already been utilized in the clinic for the healing of long bone fractures and spinal fusion surgery. However, there are many undesirable side-effects observed after BMP2 use, including increased osteolysis or bone resorption. Therefore, it is not a viable treatment of OP [11]. Further research into the BMP pathway is of interest since it is known to activate and increase both osteoblast and osteoclast activity.

Multiple discrepancies have been noted with BMP2 and the BMP signaling pathway in patients diagnosed with OP [13,14]. This increases interest in investigating the signaling pathway, not only for the development of new therapeutics, but to better understand the underlying molecular mechanisms leading to OP. Additionally, our lab has developed a novel peptide, CK2.3, that also utilizes the BMP signaling pathway to induce the SMAD-dependent and SMAD-independent signaling cascades. We have previously shown CK2.3′s efficacy in multiple cell and animal models with its pro-osteoblast and anti-osteoclast effects [20,21,22]. We have also shown that CK2.3 activates ERK and SMAD signaling in C2C12 cells [22]. Recently, we have also shown CK2.3′s increased osteogenic effect over BMP2 in cells extracted from patients diagnosed with OP when compared to control cells [23]. In fact, patients diagnosed with OP did not respond to BMP2 stimulation, but significantly responded to CK2.3 stimulation when assessed for both mineralization and osteoblast specific markers [15]. This suggests aberrant BMP2 signaling in those diagnosed with OP.

In this paper, we investigated important signaling proteins in the BMP pathway to further elucidate this signaling disparity. In particular we investigated SMAD and ERK signaling, as these are the two pathways activated by both BMP2 and CK2.3. This was completed through immunofluorescent staining and Western blotting. Activated ERK (or pERK) expression was significantly decreased after BMP2 stimulation, which indicates that SMAD independent signaling is altered in OP (Figure 1). CK2.3 has been shown to act through the ERK signaling pathway in C2C12 cells [22], and it also significantly increased expression of pERK in cells isolated from patients diagnosed with OP. This suggests that CK2.3 acts through the ERK pathway in humans as well, and that CK2.3 can rescue aberrant BMP signaling. Interestingly, SMAD-dependent signaling (or BMP canonical signaling) remained unchanged in both experiments. Even while BMP2 stimulation seemed to decrease expression of pSMAD (or activated SMAD) when compared to US and CK2.3 stimulated cells, it was not significant (Figure 2).

In the present experiments, we utilized a new explant model and verified its legitimacy by assessing cell viability (Figure 3c) and immunofluorescent validity through a confirmation study (Figure 4a,b). Increased cell viability was observed in both OP and control explants under all stimulations, as seen in Figure 1c. Both OC and ALP expression were significantly increased in explants from OP patients after CK2.3 stimulation. BMP2 stimulation significantly decreased both osteogenic biomarkers expression when compared to US. This mimics the previous in vitro study, where only CK2.3 significantly increased OC and ALP expression, while BMP2 decreased their expression. Interestingly, both BMP2 and CK2.3 significantly increased expression of OC and ALP in control patient explants when compared to US, Figure 4a,b.

BMPRIa and CK2α expression was investigated in three different models, all with the same results (Figure 5, Figure 6a,b and Figure 7a,b). Both BMPRIa and CK2α expression were significantly decreased under BMP2 stimulation in the OP models, when compared to control. BMPRIa is a critical receptor in humans within the BMP pathway. BMP2 preferentially binds to BMPRIa, and its signaling cascade is activated. Therefore, a decrease in receptor expression after BMP2 stimulation is concerning and indicates dysregulation of the BMP pathway.

In the future, more studies should be conducted that explore aberrant BMP signaling in patients diagnosed with OP. Total ERK and total SMAD levels should be investigated to determine if less total ERK or total SMAD is expressed in patients diagnosed with OP. Further investigation of the nuclear fractions of both pERK and pSMAD is of interest since nuclear regulation and mediation has been known to affect the strength and duration of a signaling event. While our lab had previously shown CK2 to interact with BMPRIa [16], another research group discovered that CK2 is a mediator of ERK phosphorylation in HeLa cells [24]. This indicates that other regulating proteins may be involved or dictate activation of the BMP signaling pathway. In the future, protein and gene expression of CK2α after stimulations should be investigated, as this interacting protein may be more involved in BMP signaling than originally thought. Additionally, histological expression of BMP2 within the femoral heads should also be investigated.

This study showed that BMP signaling is aberrant in those diagnosed with OP. BMP2 stimulation decreased activated SMAD, activated ERK, osteogenic marker expression, BMPRIa expression, and CK2α expression in OP patients’ cells or explants. Taken together, BMP2 is not a viable treatment as the signaling pathway is aberrant in OP; however, CK2.3 seemingly rescues BMP2 osteogenic mediated activity. Therefore, it has the potential to be an OP therapeutic. Further investigation of the BMP signaling pathway could lead to the development of future therapeutics to help treat those diagnosed with OP.

## 4. Materials and Methods

### 4.1. Subjects

Following institutional review board (IRB) exemption from Christiana Care Hospital, Newark, DE, USA, (10 April 2013) human femoral heads were obtained after being extracted from female patients undergoing hip arthroplasty surgery (DDD# 602228). The patients were diagnosed with osteoporosis or osteoarthritis (control). A total of 81 femoral heads were collected, of which 63 (aged 37–92) were from patients diagnosed with osteoporosis and 18 (aged 56–86) were from patients diagnosed with osteoarthritis.

### 4.2. MMA Embedding

Preserved femoral heads from patients diagnosed with either OP or osteoarthritis (OA) were aged 58–95 (11 total patients) and 41–66 (7 total patients), respectively. Using a modified embedding protocol from Akkiraju and colleagues [25], femoral heads were fixed in 10% Neutral Buffered Formalin (NBF). Once fixed the bones were cut down the midsagittal plane, and an area of trabecular bone was removed from the interior region of the bone, location shown in Figure 1b. The bone fragments were washed with 1× PBS at room temperature. They were then subsequently dehydrated using a series of ethanol dilutions. The ethanol dilution series started with a 70% ethanol incubation for 8–16 h, 90% ethanol incubation for 8–16 h, 95% ethanol incubation for 8–16 h, two changes of 100% ethanol for 8–16 h each, two changes of 100% isopropanol for 8–16 h each, and two changes of methyl salicylate for 4 h each. Once completely dehydrated samples were infiltrated with methyl methacrylate (MMA) I (750 mL MMA, 140 mL N-butyl pthalate) for 48 h at room temperature. Next, they were infiltrated with MMA II (750 mL MMA, 140 mL N-butyl pthalate, and 9 g of dry benzoyl peroxide) for 48 h at 4 °C. Last, they samples were infiltrated with MMA III (750 mL MMA, 140 mL N-butyl pthalate, 17.75 g of dry benzoyl peroxide) for 48 h at 4 °C. Samples were embedded in glass vials, and once hardened, the samples were placed in a 40 °C oven for 7 days in order to fully solidify the samples. Once fully solid, the samples were removed from the glass vials through carefully breaking the glass with a rubber mallet. They were then trimmed and sectioned using a diamond wafering blade (Buehler, Lake Bluff, IL, USA) using an IsoMet low speed Saw (Buehler, Lake Bluff, IL, USA). The sections were then sanded down using Carbimet Abrasive discs (Buehler, Lake Bluff, IL, USA) of sand paper.

### 4.3. Antigen Retrieval and Immunostaining of Embedded Samples

Cut and sanded MMA samples with approximately 200–600 µm thickness were placed in a Xylene solution for one minute to dissolve back the plastic resin. They were then placed in a prewarmed (37 °C) testicular hyaluronidase solution (47 mL 0.1 M potassium phosphate, 3 mL 0.1 M sodium phosphate, and 0.025 g testicular hyaluronidase) for 30 min. The samples were washed with 1× PBS three times following the incubation. Samples were then blocked with 3% Bovine Serum Albumin (BSA) for one hour at RT and then incubated with their designated primary antibodies overnight at 4 °C. Primary antibodies included: BMPRIa goat polyclonal IgG as a 1:200 dilution (200 µg/mL, Santa Cruz Biotechnology, Dallas, TX, USA), and CK2α (C-18) sc-6479 rabbit polyclonal IgG as a 1:200 dilution (200 μg/mL, Santa Cruz Biotechnology, Dallas, TX, USA). Following overnight incubation, the samples were washed three times in 1 × phosphate-buffered saline (PBS) for 15 min each and then incubated for the corresponding secondary antibodies for one hour at RT. Secondary antibodies include: Donkey anti goat IgG 488 (ab150129) as a 1:500 dilution (Abcam, Cambridge, UK) and chicken anti rabbit IgG 568 (A-21442) as a 1:500 dilution (Thermo Fischer, Waltham, MA, USA). The samples were washed three times with 1X PBS for 15 min each and then stained for the nucleus using Hoescht (bisbenzimide, Sigma-Aldrich, St. Louis, MO, USA, Hoechst dye No. 33258, dissolved in H_2_O) for ten minutes and subsequently washed with 1X PBS. Embedded bone slices were imaged using Zeiss LSM 710 at the 20×/0.75 Plan Apochromat objective (Flour, Zeiss, Germany). After the images were collected, the pixel intensity was determined through the “Measure” function of ImageJ (NIH, Bethesda, MD, USA).

### 4.4. Explant Culture

Femoral heads were collected within 48 h post extraction. Using nose pliers and DREMEL, 4000 trabecular bone fragments (2 mm) were extracted from the interior of the bone, as shown in Figure 1c. Once removed, the samples were washed with 1× PBS, and then placed in a 100% antibiotic/antimycotic solution for ten minutes. Following this incubation samples were placed in a six well plate with Dulbecco’s Modification of Eagles Medium (DMEM) and 10% Fetal Bovine Serum (FBS) solution. Fragments were stimulated with 40 nM of BMP2 and 100 nM of CK2.3 as designated for five days, with media and re-stimulation occurring on the third day. These specific concentrations were reported and verified to induce mineralization in a variety of cell lines and animal models [18,19,20,21,22,23].

### 4.5. Explant Viability Staining

In order to test the efficacy of utilizing a bone explant model, a cell viability assay was conducted. After the five-day stimulation period, the bone fragments were stained for viable cells using CellTrace™ Calcein Red-Orange, AM, 1 µM solution was used directly into the media. This dye was readily taken up by eukaryotic cells with a retained cell membrane, indicating that the dyed cells are viable. A Hoescht nuclear stain (1 µL of a 1:1000 dilution, directly into the media) was also used in order to determine the amount of live and dead cells present within a sample. After ten minutes, the media was aspirated, and the samples were washed once with 1× PBS. Approximately two milliliters of PBS was left with the samples while imaging. The samples were imaged using Zeiss LSM 710 at the 20×/0.75 Plan Apochromat objective (Fluor, Zeiss, Germany). The images were collected in z-stacks, ranging in size from 30–40 slices per sample for the entire sample. The images were analyzed using ImageJ (NIH, Bethesda, MD, USA) through the “Measure” function, slice by slice. Cells stained for Hoescht, Calcein, and Hoescht and Calcein were counted, slice by slice. This was completed to determine both the relative intensities of the stains and the number of cells stained per label to determine the percent cell viability throughout the entire explant sample.

### 4.6. Immunostaining Explant Cultures

After stimulation, the fragments were fixed with 4.4% Paraformaldhyde (PFA) overnight at 4 °C. Following fixation, the fragments were washed with 1× PBS five times and then the samples were blocked in 3% BSA for one hour at room temperature. The fragments were then incubated with primary antibodies overnight at 4 °C. Primary antibodies include BMPRIa (same as above) and CK2α (same as above). After primary antibody incubation the fragments were washed in 1× PBS three times for 15 min each. They were then incubated with secondary antibodies (same as above) for one hour at room temperature. After that incubation, the fragments were washed with three changes of 1× PBS for 15 min each. Approximately 1 mL of 1 × PBS was aliquoted into the wells with the explants and the explants were imaged using Zeiss LSM 710 with the 20×/0.70 W Plan Apochromat objective (Fluor, Zeiss, Germany). Images were collected as z-stacks, averaging approximately 30–40 slices per sample for the entire sample. Images were analyzed using the “Measure” function of ImageJ (NIH, Bethesda, MD, USA), slice by slice, in order to obtain the relative pixel intensities of each of the aforementioned stains. Pixel intensities were averaged for each stack and for each patient. This was completed in three OP patients and three control patients.

### 4.7. Isolation of Primary Osteoblasts

Primary osteoblasts were isolated as previously described [15]. Briefly, femoral heads were obtained and in a sterile environment, femoral heads sliced open and trabecular bone fragments were collected from the interior surface of the bone, washed with 1× PBS, and digested with a DMEM/collagenase (Corning; Collagenase Type II, Worthington, Columbus, OH, USA) solution for two days. The cellular suspension was filtered, resuspended in fresh DMEM, and plated in a T25 flask. Cells were cultured every four to seven days until reaching 90% confluency.

### 4.8. Immunostaining Osteoblasts

Cells were isolated from five female osteoporotic patients whose ages were 60–92. The cells were seeded in a 24-well plate at a density of 1 × 10^5^ cells/cm^2^ per well. Once 90% confluent cells were serum-starved overnight and treated with 100 nm CK2.3 or 40 nm BMP2 or left unstimulated (US). After five days of treatment, cells were washed with 1× PBS and then fixed with 4% (*w*/*v*) Paraformaldehyde (PFA) for 15 min at room temperature. They were then permeabilized using 1% (*w*/*v*) saponin for 10 min on ice. The samples were fluorescently labeled for one hour at room temperature with their corresponding primary antibody. Primary antibodies used: BMPRIa (same as above), CK2α (same as above), pERK E-4 (sc-7383) (mouse monoclonal antibody) as a 1:1000 dilution, 200 ug/mL, Santa Cruz Biotechnology, Dallas, TX, USA), and pSMAD1/5 Ser 463/465 (41D10) (phosphoSMAD1/5 rabbit monoclonal antibody) as a 1:1000 dilution (Cell Signaling, Danvers, MA, USA). This incubation was followed by another hour incubation with the corresponding secondary antibodies at room temperature. They included: Alexafluor 488 chicken antirabbit IgG (A-21441) as a 1:500 dilution (200 µg/mL, Thermo Fischer, Waltham, MA, USA), Alexafluor 568 donkey antigoat IgG (ab177454) as a 1:500 dilution (200 µg/mL, Abcam, Cambridge, UK). All antibodies were diluted in a 3% bovine serum albumin (BSA) solution. Cells were stained for their nucleus using Hoescht for two and a half minutes. The coverslips were mounted using Airvol, as previously described. Images were taken on Zeiss Axiophot with a 20× objective (Fluor, Zeiss, Germany) and analyzed in ImageJ (NIH, Bethesda, MD, USA).

### 4.9. Imaging Quantification

#### 4.9.1. MMA and Osteoblast Immunostaining

Immunofluorescent images were quantified using ImageJ. Briefly, images were converted to 8 bits, and the threshold was then adjusted to the 2nd or negative control to eliminate nonspecific staining. Once converted, the images were black and white, which made it easier to calculate pixel intensity. Pixel intensity was calculated through the measure function of ImageJ and was averaged for both control and OP bone slices. Fluorescent staining intensity has been shown to be equivalent to the pixel intensity measured in ImageJ [26].

#### 4.9.2. Explant Live/Dead Analysis

Images were collected in z-stacks and were quantified through ImageJ. Image stacks averaged about 40 slices per stack, and each slice was analyzed for the red fluorescent intensity, representing viable cells stained for Calcein. Individual cells were also counted in each slice that were stained red for viability and blue for the nucleus. Cells stained with both blue and red were indicative of live and viable cells, whereas cells just stained blue were indicative of dead cells. The viable cell population was calculated for each stimulation under each sample type.

#### 4.9.3. Explant Immunostaining

Images were again collected in z-stacks and were quantified through ImageJ. Image stacks averaged 40 slices per stack. Each slice was analyzed for fluorescent intensity through the measure function of ImageJ. Readings were averaged and analyzed for each sample type as well as for each stimulation. Negative or secondary control readings were also measured and subtracted from the averaged readings.

### 4.10. Lysate Collection

Cells were plated on a 6 well plate at a density of 1 × 10^5^ cells/cm^2^. Once 90% confluent, the cells were serum starved overnight and treated with either 40 nM BMP2 and 100 nM CK2.3 or left unstimulated (US) for five days. On the fifth day, the cells were washed with ice cold IX PBS and incubated with lysis buffer (containing 10 mM Tris pH 7.5, 50 mM NaCl, 1% Triton X-100, 60 mM octyl glucoside, 1 mM PMSF, 10 mg/mL each of leupeptin, aprotinin, soybean trypsin inhibitor, benzamidine-HCl, pepstatin, and antipain) for 1 h, as previously described [27].Cells were then sonicated (30 s, two times) and centrifuged at 14,000× *g* for 20 min in order to remove cellular debris. Protein concentrations were determined using a Promega Glomax plate reader following manufacturers protocols (Pierce™ BCA protein Assay Kit, Thermo Fischer, Waltham, MA, USA).

### 4.11. Western Blot

Once protein concentration was determined, samples were normalized and loaded into a 12.5% SDS-Polyacrylamide gel. The gel was run for 90 min at 90 V and then the protein extracts were transferred onto presoaked PVDF membrane for one hour at 15 V using a semi-dry transfer (BioRad, Hercules, CA, USA). Once the protein had fully transferred, the membrane was blocked using 5% BSA in IX Phosphate Buffered Saline with 1% Tween (PBST) solution for one hour at room temperature. The membrane was then incubated with primary antibodies overnight at 4°C. Primary antibodies included: pSMAD 1/5 (same as above), pERK (same as above), and β-actin (7D2C10) (Rabbit polyclonal IgG, Proteintech, Chicago, IL, USA). The membrane was then washed three times with 1X PBST for 15 min each. It was then incubated with secondary antibodies conjugated to horseradish peroxidase (HRP) for one hour at room temperature. Secondary antibodies included: Goat anti rabbit IgG-HRP (ab6721) (Abcam, Cambridge, UK) and rabbit anti mouse IgG-HRP (ab6728) (Abcam, Cambridge, UK). The membrane was then washed three times with 1× PBST for 15 min each. The membrane was then incubated with Chemiluminescence FemtoMAX™ Super Sensitive HRP Substrate (Rockland, Gilbertsville, PA, USA) for two and a half minutes. Chemiluminescence was detected using a ChemiDoc (BioRad, Hercules, CA, USA).

### 4.12. RNA Collection

RNA was collected from three female OP patients aged 76–92. Cells were plated in a six well plate at a density of 1 × 10^5^ cells/cm^2^. Once 90% confluent, the cells were serum starved overnight and treated with either 40 nM BMP2, 100 nM CK2.3, or left US for five days. On the fifth day, the cells were washed with 1× PBS and 300 µL of TRIzol™ (Invitrogen, Carlsbad, CA, USA) solution was added to each well. The lysate was pipetted up and down in the solution and then aliquoted into fresh centrifuge tubes. Chloroform (Fischer Scientific, Waltham, MA, USA) was added to each tube, mixed well, and then incubated for ten minutes at RT. The samples were centrifuged for 15 min at 12,000× *g* at 4 °C, which separates the solution into two phases: a top aqueous phase containing the RNA and a bottom phenol/chloroform phase. The top aqueous phase was removed and aliquoted into fresh centrifuge tubes. Isopropanol (Fischer Scientific, Waltham, MA, USA) was added to each tube, and incubated for ten minutes at RT. They were then centrifuged for ten minutes at 12,000× *g* at 4 °C, which precipitates the RNA into a pellet. The supernatant was removed, and the pellet was washed two times with 75% Ethanol (Fischer Scientific, Waltham, MA, USA), vortexing between washes, and the pellet was then air dried for ten minutes at RT. The RNA was then re-suspended in 100 µL of RNase free water (Fischer Scientific, Waltham, MA, USA).

### 4.13. Quantitative Reverse Transcription Polymerase Chain Reaction

Two step RT-PCR was performed with 2 µg of RNA obtained and using a high capacity cDNA reverse transcription kit. In the second step the obtained cDNA was amplified through PCR using specific primers (Integrated DNA Technologies, Coralville, IA, USA). The primer sequences used are as follows (1) *BMPRIa* (forward) *CAG CCT CCA GAC TCA CAG CAT* (reverse) *CGA GAC CCA TGA CTT AAG G. GAPDH* was used as the housekeeping gene, its primer sequences were (forward) *CAT GGC CTT CCG TGT TCC TA* (reverse) *CCT GCT TCA CCA CCT TCT TGA T*. The primer sequences have been used and verified in several publications [13,28,29]. The RT-qPCR used Fast SYBR™ Green Master Mix per the manufacturer’s protocols (Thermo Fischer, Waltham, MA, USA). Fold change in mRNA expression was processed using procedures outlined previously [30].

### 4.14. Statistical Analysis

Data were analyzed through an ANOVA with a Tukey–Kramer post hoc test. Outliers were removed through Chauvenet’s criterion, and error bars depict standard error of the mean.

## Figures and Tables

**Figure 1 ijms-21-06909-f001:**
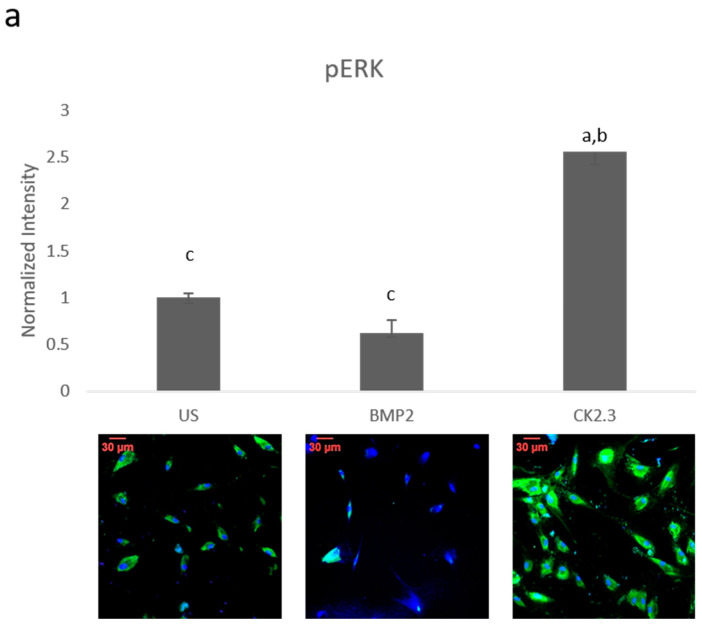
Mature osteoblasts were extracted from osteoporosis (OP) patients and stimulated with BMP2, CK2.3 or were left unstimulated (US) for five days. On the fifth day cells were fixed and fluorescently stained for pERK (green) and the nucleus (blue). (**a**) CK2.3 significantly increased fluorescent intensity when compared to BMP2 and US cells. BMP2 stimulation significantly decreased fluorescent intensity when compared to CK2.3 stimulated cells. (**b**) Lysates were also collected from extracted osteoblasts from OP patients and run on an SDS-Page gel to separate the proteins. The separated proteins were then transferred onto an immunoblot and pERK and β-actin protein levels were detected. Expression was detected through densiometric analysis using ImageJ. CK2.3 significantly increased expression of pERK when compared with US and BMP2 stimulated cells. Error bars represent standard error of the mean (SEM). “a” denotes statistically significant to US, “b” denotes statistically significant to BMP2 stimulated cells, and “c” denotes statistically significant to CK2.3 stimulated cells (*p* < 0.05).

**Figure 2 ijms-21-06909-f002:**
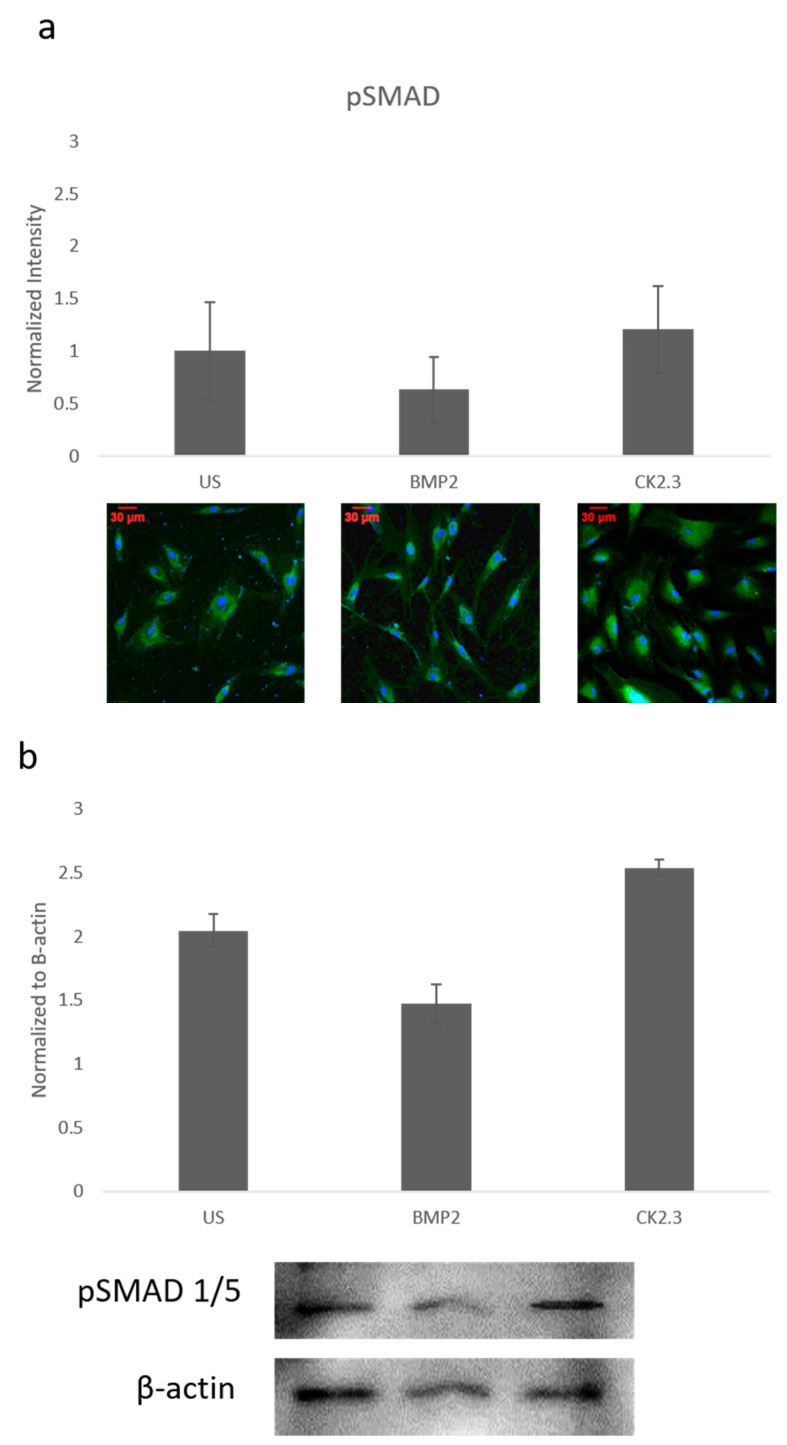
Mature osteoblasts were extracted from OP patients and stimulated with BMP2, CK2.3 or were left US for five days. On the fifth day, cells were fixed and fluorescently stained for pSMAD (green) and the nucleus (blue). (**a**) pSMAD fluorescent intensity remains unchanged under all stimulations, while BMP2 stimulation indicates a slight decrease in expression, this was not significant. (**b**) Lysates were also collected from extract osteoblasts from OP patients and run on an SDS-Page gel to separate the proteins. The separated proteins were then transferred onto an immunoblot and pSMAD and β-actin protein levels were detected. Expression was detected through densiometric analysis. BMP2 stimulation seems to decrease pSMAD expression, but this difference was not significant. Error bars represent standard error of the mean (SEM). (*p* < 0.05).

**Figure 3 ijms-21-06909-f003:**
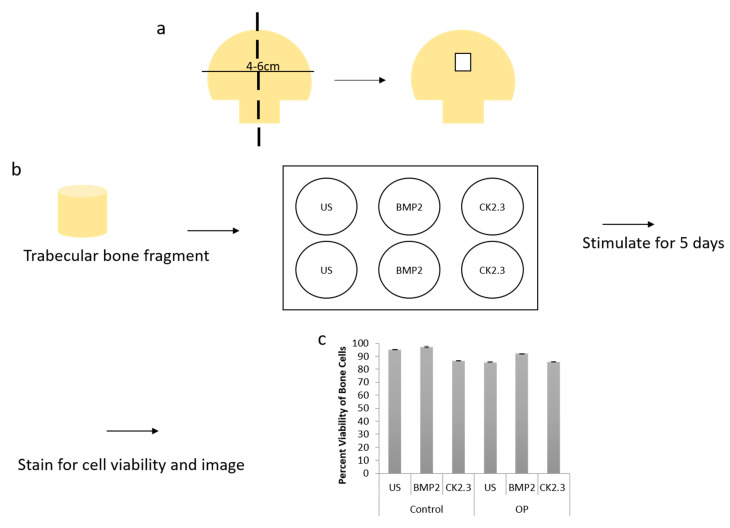
(**a**) A schematic representing how femoral heads were sliced down the midsagittal plane, and what region of interest was used for both the methyl methacrylate (MMA) experiments and the explant experiments. (**b**) Diagram showing the explant experimental set up. The trabecular bone fragment was removed from the femoral head, washed with PBS incubated with antibiotics/antimycotics for 10 min. The fragments were placed in a six well plate (one fragment per well) with DMEM. They were stimulated as designated for five days, following which they were either stained for cell viability and imaged or they were fixed with 4.4% PFA, immunofluorescently stained and then imaged. (**c**) Cell viability was assessed through a Calcein and Hoescht stain, and viable cells were counted. Under all stimulations and conditions cells were 80% or more viable within the explants. Error bars represent standard error of the mean (SEM).

**Figure 4 ijms-21-06909-f004:**
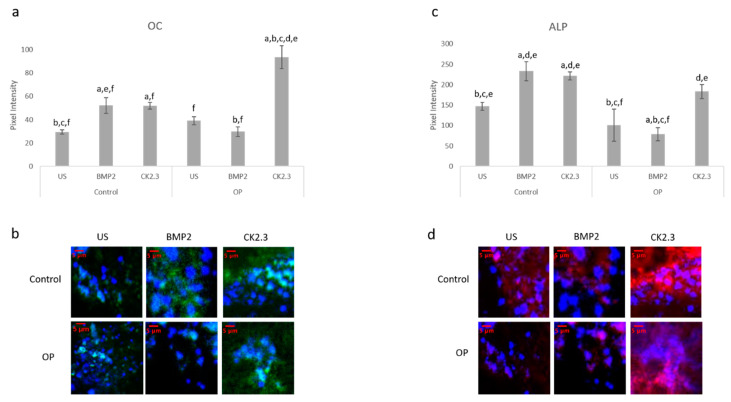
Explants from OP and control patients were prepared as previously described and stimulated with BMP2, CK2.3, or left US. After the fifth day the explants were fixed and stained for OC (green), ALP (red), and the nucleus (blue). (**a**) Control and OP explants stimulated with CK2.3 significantly increased expression of OC when compared to US and BMP2 stimulation. OP explants stimulated with BMP2 significantly decreased expression of OC. (**b**) Representative 2D images depicting the nuclear stain overlayed with the OC stain. (**c**) Both control and OP explants stimulated with CK2.3 significantly increased expression of ALP when compared to control and BMP2 stimulation. BMP2 stimulation significantly decreased ALP expression in both control and OP explants. (**d**) Representative 2D images depicting the nuclear stain overlaid with the ALP stain. Error bars represent standard error of the mean (SEM). “a” denotes statistically significant to control US explants, “b” denotes statistically significant to control BMP2 stimulated explants, “c” denotes statistically significant to control CK2.3 stimulated explants, “d” denotes statistically significant to OP US explants, “e” denotes statistically significant to OP BMP2 stimulated explants, and “f” denotes statistically significant to OP CK2.3 stimulated explants (*p* < 0.05).

**Figure 5 ijms-21-06909-f005:**
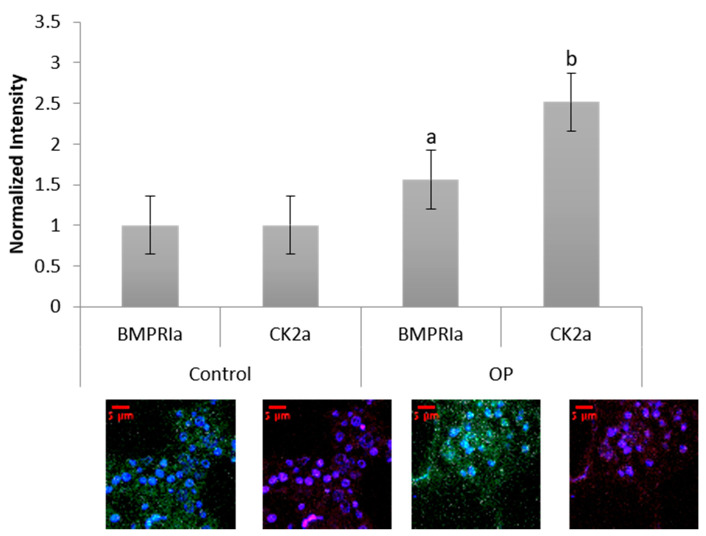
Female OP and OA (control) trabecular bone slices were embedded in MMA and fluorescently stained for BMPRIa (green), CK2α (red), and the nucleus (blue). OP BMPRIa and CK2α expression was significantly higher than the expression levels in control bone slices. Error bars represent standard error of the mean (SEM). “a” denotes statistically significant to control BMPRIa levels, “b” denotes statistically significant control CK2α levels (*p* < 0.05).

**Figure 6 ijms-21-06909-f006:**
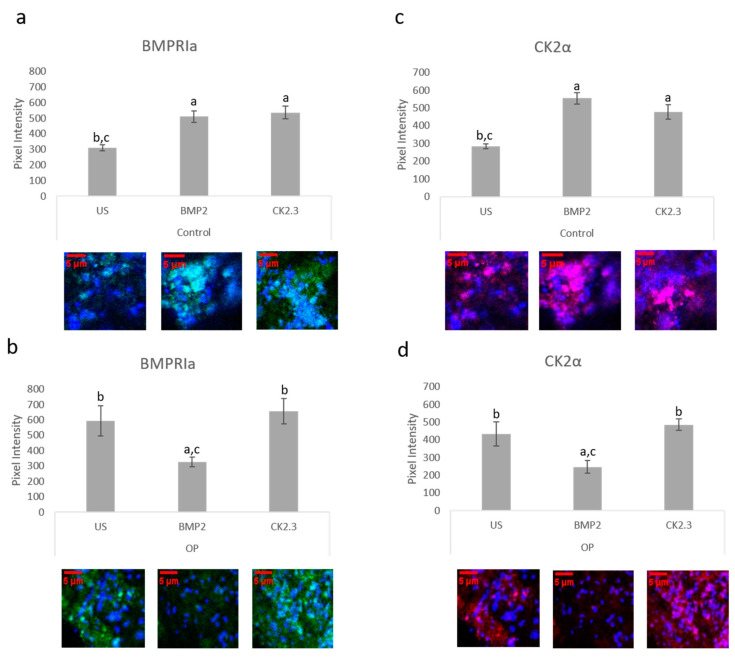
Trabecular bone slices were removed from isolated femoral heads. OP and control explants were prepared as previously described. Fixed explants were stained fluorescently for BMPRIa (green), CK2α (red), and the nucleus (blue). (**a**) In control explants CK2.3 and BMP2 significantly increased BMPRIa expression when compared to US explants. (**b**) In OP explants, BMP2 significantly decreased BMPRIa expression when compared to CK2.3 and US explants. (**c**) In control explants, BMP2 and CK2.3 significantly increased expression of CK2α when compared to US explants. (**d**) In OP explants, BMP2 stimulation significantly decreased expression of CK2α when compared to US and CK2.3 stimulated explants. Error bars represent standard error of the mean (SEM). “a” denotes statistically significant to US explants, “b” denotes statistically significant to BMP2 stimulated explants, and “c” denotes statistically significant to CK2.3 stimulated explants (*p* < 0.05).

**Figure 7 ijms-21-06909-f007:**
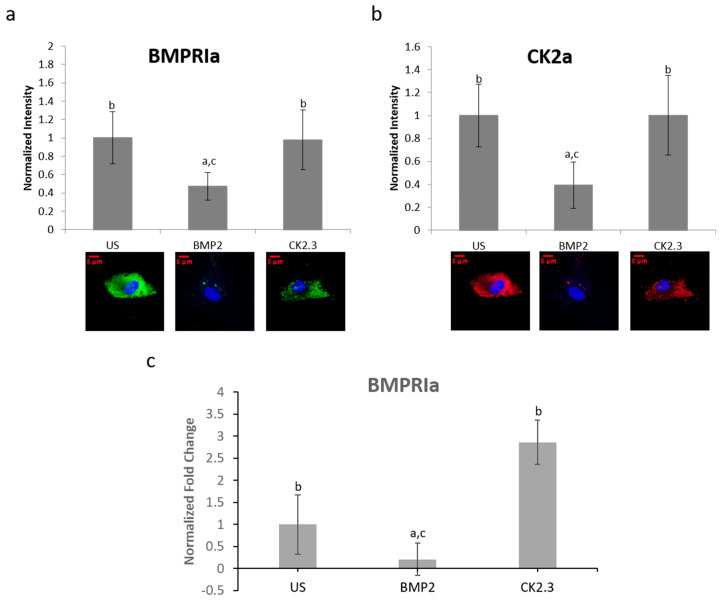
Osteoblasts extracted from OP patients were stimulated with BMP2, CK2.3, or left US for five days. After the fifth day, the cells were fixed and stained for BMPRIa (green), CK2α (red), and the nucleus (blue). (**a**) CK2.3 and US cells significantly increased expression of BMPRIa when compared to BMP2 stimulation. (**b**) BMP2 stimulation significantly decreased expression of CK2α when compared to CK2.3 and US cells. (**c**) RNA was extracted from OP patients’ osteoblasts after they were stimulated with BMP2, CK2.3, or left US for five days. BMP2 significantly decreased *BMPRIa* gene expression when compared to US and CK2.3 stimulated cells. Error bars represent standard error of the mean (SEM). “a” denotes statistically significant to US, “b” denotes statistically significant to BMP2, and “c” denotes statistically significant to CK2.3 (*p* < 0.05).

**Figure 8 ijms-21-06909-f008:**
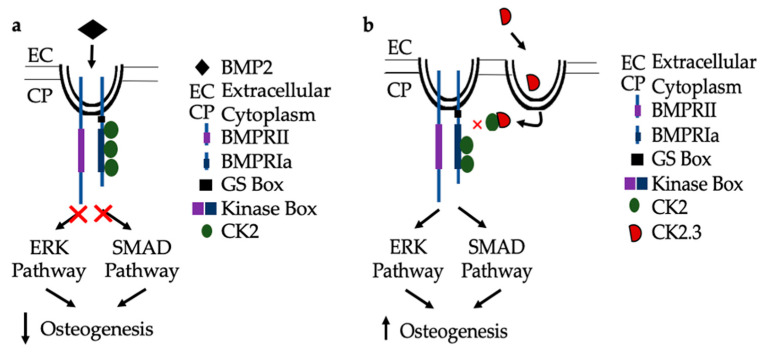
Proposed mechanism of BMP2 (**a**) and CK2.3 (**b**) signaling in POP primary osteoblasts. (**a**) Upon treatment with BMP2, POP primary osteoblasts are unresponsive and the downstream signaling pathways, ERK and SMAD, are not activated and osteogenesis is decreased. (**b**) When stimulated with CK2.3, this peptide is up-taken by POP primary osteoblasts and subsequently, the ERK and SMAD signaling pathways are activated independently of BMP2, which increases osteogenesis.

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
