# Peer review of "Aberrant BMP2 Signaling in Patients Diagnosed with Osteoporosis"

_ijms, 2020, doi:10.3390/ijms21186909_

Round 1

Reviewer 1 Report

The paper studies the BMP2 role in osteoporosis. present paper propose and use a novel strategy for experimental study of BMP2 pathway in osteoporosis. IN my opinion, there are too many pictures and information for Results section. I propose to filter the images and to reduce them in number. Although, I would like to replace one of it with a histological section from human bone when it may use BMP2 immunohistochemical expression of BMP2 in normal bone and osteoporotic bone.

References must be updated with the title mostly from 208-2020.

Author Response

The authors would like to thank the reviewer for the extensive and thoughtful feedback. The insightful comments and suggested edits greatly improved the manuscript. Below, please find our point-by-point revision to each critique and/or question suggested by the reviewer. All changes in the manuscript can be seen by using the “Track Changes” plug-in of Microsoft word.

Point 1: IN my opinion, there are too many pictures and information for Results section. I propose to filter the images and to reduce them in number. Although, I would like to replace one of it with a histological section from human bone when it may use BMP2 immunohistochemical expression of BMP2 in normal bone and osteoporotic bone.

Response 1: We thank the reviewer for their kind feedback. We believe the order and layout of the results make it easier for the reader to follow the research story. The reviewer also suggested to include a histological cross-section of human bone stained for BMP2 expression, which was an excellent suggestion. We have included this thought in the future directions portion of the discussion.

Point 2: References must be updated with the title mostly from 208-2020.

Response 2: We thank the reviewer for this feedback, the references have been updated to be within the years of 2008-2020.

Reviewer 2 Report

The authors should consider the followings:

  1. The authors should give details of the vehicle used in CK2.3, and/or BMP2. Please explain whether vehicle controls were used across the experiment if the vehicle were present.
  2. In Fig 1a, the author should increase the resolution of the microscopic images.
  3. In Fig 1a and Fig 1b, please describe the abbreviation of "a, b, c" in the respective figure legend.
  4. In the treatment, please give rationales of using the respective specific dosage,100 nM CK2.3, or 40nM BMP2?
  5. In Fig 1b, please also show the total protein of ERK1/2 (also known as tERK 1/2), in the western blotting.
  6. Please give rationale(s) for using beta-actin as housekeeping protein in this experiment setting.
  7. The authors are encouraged to specify whether the error bars were in S.D. or S.E.M. in the individual figure legend, and that for cell line studies, the use of S.D. in error bars are recommended.
  8. In Fig 2a, the quantitative graphs were not exactly reflecting the representative micrographs. Please select the most representaive micrographs in the display of results. Otherwise, please explain this discrepancy.
  9. In Fig 2b, please show the SMAD1/5 protein (together with pSMAD1/5) in the same samples. Please otherwise explain why only "beta-actin" was used instead of together with SMAD1/5.
  10. In the signaling pathway studies of pERK and pSMAD1/5, the nuclear fraction were also frequently included. Please explain why the authors did not include such part of the studies.
  11. Did the authors reproduce the results (described in Figure 4, and in Figure 5) in similar trend using other system (such as in western protein expression for OC and ALP), or gene expression studies?
  12. Please explain why OA was used as control.
  13. In Figure 7c, please give rationale(s) to the choice of housekeeping used in the study. What were the changes in gene expression for CK2a?
  14. The authors may consider using English proof-reading services by language professional.
  15. Please state clearly the novelty of this research in your abstract and conclusion.
  16. Please include the clone number (or specific ID) for the antibodies used in the studies, for traceability purposes.

Author Response

The authors would like to thank the reviewer for the extensive and thoughtful feedback. The insightful comments and suggested edits greatly improved the manuscript. Below, please find our point-by-point revision to each critique and/or question suggested by the reviewer. All changes in the manuscript can be seen by using the “Track Changes” plug-in of Microsoft word.

The authors should consider the followings:

Point 1. The authors should give details of the vehicle used in CK2.3, and/or BMP2. Please explain whether vehicle controls were used across the experiment if the vehicle were present.

Response 1: We thank the reviewer for pointing this out. The vehicle that was used for both CK2.3 and BMP2 delivery was sterile water and because of this, no vehicle controls were needed or used.

Point 2: In Fig 1a, the author should increase the resolution of the microscopic images.

Response 2: The resolution on the microscope images was improved and the figure has been replaced.

Point 3: In Fig 1a and Fig 1b, please describe the abbreviation of "a, b, c" in the respective figure legend.

Response 3: We thank the reviewer for pointing out this discrepancy, we have added in the meaning of “a”, “b”, and “c” into the figure legend in Figure 1. Briefly, those letters refer to the statistically significant designations between variables.

Point 4: In the treatment, please give rationales of using the respective specific dosage,100 nM CK2.3, or 40nM BMP2?

Response 4: The reviewer brings up an excellent point, which should have been made clearer in the text.  The CK2.3 and BMP2 concentrations used were selected because our lab has previously shown that 100nm of CK2.3 and 40nm of BMP2 were the optimal, in vitro concentrations to produce an osteogenic response. Previously, concentration curves were conducted in C2C12 cells and from there the optimal mineralization inducing concentrations were determined. This detail has been added into the methods section of the paper.

Point 5: In Fig 1b, please also show the total protein of ERK1/2 (also known as tERK 1/2), in the western blotting.

Response 5: We appreciate the feedback from the reviewer. We believe tERK is not needed because we are using a control (Beta-actin) to directly compare to pERK levels in stimulated cells. However, we recognize the importance of looking into tERK levels as compared with pERK levels, and we propose this in the future studies section of the discussion.

Point 6: Please give rationale(s) for using beta-actin as housekeeping protein in this experiment setting.

Response 6: We appreciate the reviewer’s thoughtful comment. Beta-actin is used as the housekeeping protein because within both C2C12 cells and patient cells, this protein is commonly expressed. This common expression is retained in both the stimulated and unstimulated conditions. Therefore, due to its high expression levels and expression in all cell types and conditions, beta-actin gives us a reliable comparison for the other bands.

Point 7: The authors are encouraged to specify whether the error bars were in S.D. or S.E.M. in the individual figure legend, and that for cell line studies, the use of S.D. in error bars are recommended.

Response 7: All error bars throughout the paper represents standard error of the mean (SEM). This has been added at the end of each figure legend.

Point 8: In Fig 2a, the quantitative graphs were not exactly reflecting the representative micrographs. Please select the most representaive micrographs in the display of results. Otherwise, please explain this discrepancy.

Response 8: We thank the reviewer for pointing out this discrepancy. The micrographs have been replaced in order to show more representative images and reflect the graph.

Point 9: In Fig 2b, please show the SMAD1/5 protein (together with pSMAD1/5) in the same samples. Please otherwise explain why only "beta-actin" was used instead of together with SMAD1/5.

Response 9: Like point 5, looking into SMAD 1/5 may be redundant as we are specifically looking into activation. Beta-actin is used because regardless of stimulation, the concentration of this protein will be similar if the cell concentration is similar, thus indicating any changes in protein expression. However, it is again important to note that further research into both tSMAD and tERK is needed to determine if lower amounts of tERK and tSMAD is affecting the overall signaling cascade. This was added into the future directions section of the discussion.

Point 10: In the signaling pathway studies of pERK and pSMAD1/5, the nuclear fraction were also frequently included. Please explain why the authors did not include such part of the studies.

Response 10: We thank the reviewer for mentioning this topic. We chose not to investigate or include pERK and pSMAD in the nuclear fraction because this was a preliminary study. We first wanted to investigate whether or not BMP signaling was activated in patients diagnosed with osteoporosis. However, moving forward, investigating the nuclear fraction and fate or both pERK and pSMAD is of interest. We included this in the future studies portion of the discussion.

Point 11: Did the authors reproduce the results (described in Figure 4, and in Figure 5) in similar trend using other system (such as in western protein expression for OC and ALP), or gene expression studies?

Response 11: We had previously conducted an immunofluorescent study on isolated osteoblasts from patients diagnosed with osteoporosis. This was conducted after the cells were stimulated with BMP2, CK2.3 or left US (the same as Figure 4). In that study, (and this current study), we had observed the same trend in OC and ALP expression following stimulations. In Figure 4, we used an explant model, and we include control explants as a comparison.

Point 12: Please explain why OA was used as control.

Response 12: OA was used as a control because cells and explants isolated from OA patients’ femoral heads positively responded to BMP2 stimulation. In fact, OA patients’ cells and explants responded the same way as C2C12 cells. We were unable to obtain normal femoral heads from Christiana Care, as all patients in our sample underwent hip arthroplasty were either diagnosed with OP or OA.

Point 13: In Figure 7c, please give rationale(s) to the choice of housekeeping used in the study. What were the changes in gene expression for CK2a?

Response 13: We thank the reviewer for pointing this out. We used GAPDH as our housekeeping gene in this study, as this gene is highly expressed in all cell types. The normalized fold change in 7c is referring to the difference between BMPRIa and GAPDH gene expression, and then this value was normalized. We were unable to obtain gene expression for CK2a, but this is something we hope to investigate in the future and has been included in the discussion.

Point 14: The authors may consider using English proof-reading services by language professional.

Response 14: We thank the reviewer for noting some spelling and grammar issues. We have thoroughly proofread and corrected these issues throughout the paper.

Point 15: Please state clearly the novelty of this research in your abstract and conclusion.

Response 15: This research is novel because it shows that there is a discrepancy in the BMP signaling pathway in those diagnosed with osteoporosis. Further investigation of the BMP signaling pathway could lead to the development of future therapeutics to help treat those diagnosed (or will be diagnosed) with osteoporosis. This has been added into both the abstract and the conclusion.

Point 16: Please include the clone number (or specific ID) for the antibodies used in the studies, for traceability purposes.

Response 16: We thank the reviewer for pointing out this missing detail. The clone number or specific ID was included for all antibodies in the methods section.
